# Covalent Immobilization of Proteases on Polylactic Acid for Proteins Hydrolysis and Waste Biomass Protein Content Valorization

Eleonora Calzoni [1,†], Alessio Cesaretti [1,†], Silvia Tacchi [2], Silvia Caponi [2], Roberto Maria Pellegrino [1], Francesca Luzi [3], Francesco Cottone [4], Daniele Fioretto [4,5], Carla Emiliani [1,5] and Alessandro Di Michele [4,*]

[1] Department of Chemistry, Biology and Biotechnology, University of Perugia, 06123 Perugia, Italy; eleonoracalzoni@gmail.com (E.C.); alex.cesaretti14@gmail.com (A.C.); roberto.pellegrino@unipg.it (R.M.P.); carla.emiliani@unipg.it (C.E.)
[2] Department of Physics and Geology, Istituto Officina dei Materiali of the CNR (CNR-IOM)—Unit of Perugia, University of Perugia, 06123 Perugia, Italy; tacchi@iom.cnr.it (S.T.); silvia.caponi@cnr.it (S.C.)
[3] Civil and Environmental Engineering Department, University of Perugia, UdR INSTM, Strada di Pentima 4, 05100 Terni, Italy; francesca.luzi@unipg.it
[4] Department of Physics and Geology, University of Perugia, 06123 Perugia, Italy; francesco.cottone@unipg.it (F.C.); daniele.fioretto@unipg.it (D.F.)
[5] Centro di Eccellenza sui Materiali Innovativi Nanostrutturati-CEMIN, University of Perugia, 06123 Perugia, Italy
* Correspondence: alessandro.dimichele@collaboratoti.unipg.it
† Both authors contributed equally to this work.

**Abstract:** The recovery of the protein component and its transformation into protein hydrolysates, generally carried out chemically, gives great added value to waste biomasses. The production of protein hydrolysates through enzymatic catalysis would guarantee to lower the environmental impact of the process and raise product quality, due to the reproducible formation of low molecular weight peptides, with interesting and often unexplored biological activities. The immobilization of the enzymes represents a good choice in terms of stability, recyclability and reduction of costs. In this context, we covalently linked proteases from *Aspergillus oryzae* to polylactic acid an eco-friendly biopolymer. The hydrolytic efficiency of immobilized enzymes was assessed testing their stability to temperature and over time, and checking the hydrolysis of model biomasses (casein and bovine serum albumin). Soybean waste extracts were also used as proof of principle.

**Keywords:** biocatalysis; enzyme immobilization; degradation of agro-food biomasses; biomaterials; SEM; AFM; Micro Raman

## 1. Introduction

The use of enzymes in the industrial field has grown enormously in recent years. By virtue of their high catalytic power, specificity and ability to work under mild conditions of temperature and pH, enzymes can be used to regulate various kinds of processes [1,2]. To date, these biomolecules are used in multiple industrial fields, from the textile to the food industry, and from the pharmaceutical sector to the production of biofuels [3], where they have supplanted the use of inorganic catalysts. The latter, although usable for a larger number of operating cycles, are chemical species that often operate under severe conditions of temperature and pH; hence, the alternative use of enzymes as biocatalysts has allowed production costs to be reduced in many processes. Another important aspect is that enzymes are biomolecules, which do not lead to the formation of toxic/pathogenic or polluting compounds, and for this reason they are considered as low environmental impact molecules. Enzymes for industrial applications can be extracted from animal, plant or microorganisms (i.e., fungi or bacteria); the choice of the species to be used must also take into account the type of industrial process in which the enzyme will be employed. Broadly

speaking, in recent years we have witnessed a growing interest in microbial sources [4]; microorganisms indeed represent an important source of enzymes by virtue of their biochemical diversity, for the exponential growth they can achieve and for their susceptibility to genetic manipulation. Furthermore, the extraction of enzymes from microbial sources is extremely cheap, as they are secreted directly in the culture medium and therefore, no excessive purification steps are required. Due to these characteristics, enzymes derived from microbial sources offer a myriad of application possibilities in the industry, guaranteeing functionality even in those sectors where plant- and animal-derived enzymes have not worked satisfactorily [5–8]. To this respect, proteases, or peptidases, constitute the largest group of enzymes used in the industrial field, representing about 60% of the enzymes employed, particularly in the detergent, food and pharmaceutical industries [9–12]. These enzymes are able to hydrolyze peptide bonds in proteins, converting them into small peptides and free amino acids. A great variety of species belongs to this class of enzymes, each with its specificity of action and function, thus making proteases extremely attractive for their multiple biotechnological applications [13,14]. However, the use of large-scale soluble proteases is problematic due to poor stability in their repeated use in batch processes. An effective strategy to be adopted is, therefore, their immobilization on inert materials. In fact, immobilized enzymes provide better resistance under different reaction conditions, allow the recovery of the product, which comes in a different phase with respect to the immobilized enzyme (heterogeneous catalysis), and cut down the costs because of their potential repeated use. The immobilization turns the enzyme from its soluble into its insolubilized form, which represents a good choice in terms of stability, recyclability and, thus, reduction of costs [15,16]. The use of immobilized enzymes in the industrial field is taking hold in many application sectors such as the environmental one for the treatment of biomasses. The degradation of biomasses derived from agriculture and food industry exhibits the double advantage of eliminating polluting wastes and introducing novel bio-derived products into the market. The recovery of these materials provides environmental and socio-economic benefits, inasmuch as the problems concerning their disposal are reduced and they can be exploited as alternative energy sources. Resorting to proteases represents a valuable means for the degradation of biomasses reach in proteins and the production of high value-added products, known as protein hydrolysate, which can be reused by virtue of their high bio-stimulating, hormonal and/or fertilizing capacities [5].

The stability and functioning of some proteases for the production of protein hydrolysates have already been tested after their immobilization on different types of materials [16,17]. For example, by means of the co-immobilization of alcalase and trypsin on composite supports of calcium-chitosan alginate, peptides have been obtained from corn zein [18]; while the immobilization of alkaline proteases on amino-functionalized magnetic nanoparticles has made it possible to obtain the hydrolysis of soy proteins more efficiently than through soluble enzymes [19]. In this light, the aim of our study is to provide and test an effective protocol for the immobilization of proteases from *Aspergillus oryzae* to be used for the degradation of proteins in waste biomass. In this study, enzymes were covalently attached to an eco-friendly material, i.e., polylactic acid (PLA) films, prepared by spin coating, where the covalent bond between enzymes and PLA was promoted by functionalization of the polymer with amine and glutaraldehyde molecules. PLA films, before and after immobilization of proteases, were characterized by thermoanalytical (differential scanning calorimetry (DSC)), microscopic (AFM and SEM) and spectroscopic (Raman) techniques. Moreover, the thermal stability of the immobilized enzymes was tested, as well as their hydrolytic capacity toward simple model biomasses. In particular, the ability to hydrolyze casein by the same enzyme–matrix system was probed for repeated cycles, whereas the capacity to degrade bovine serum albumin (BSA) was investigated as a function of incubation time. Finally, the efficacy of our system was tested towards soybean waste extracts used as model waste biomass. In fact, soybean seed is one of the most important protein sources for humans and livestock, with its annual production (for oil and animal feed) being constantly increasing, to the point that it comes fourth among

major grain crops [20]. By performing this study, the possibility to efficiently hydrolyze the protein content in waste biomasses by means of fungus-derived proteases covalently immobilized on PLA films was successfully demonstrated.

## 2. Results and Discussion

### 2.1. SEM and AFM Analyses

SEM and AFM analyses were carried out to check the binding of the protease on the activated PLA films.

Figure 1 shows SEM analyses. The morphology of the PLA film is visible in Figure 1a, showing a heterogeneous surface. After the activation process (Figure 1b) a more homogeneous surface can be observed, with irregular aggregates with sizes ranging between 20 and 40 nm. After protease immobilization, the morphology changes considerably (Figure 1c) with the development of a monolayer of enzymes.

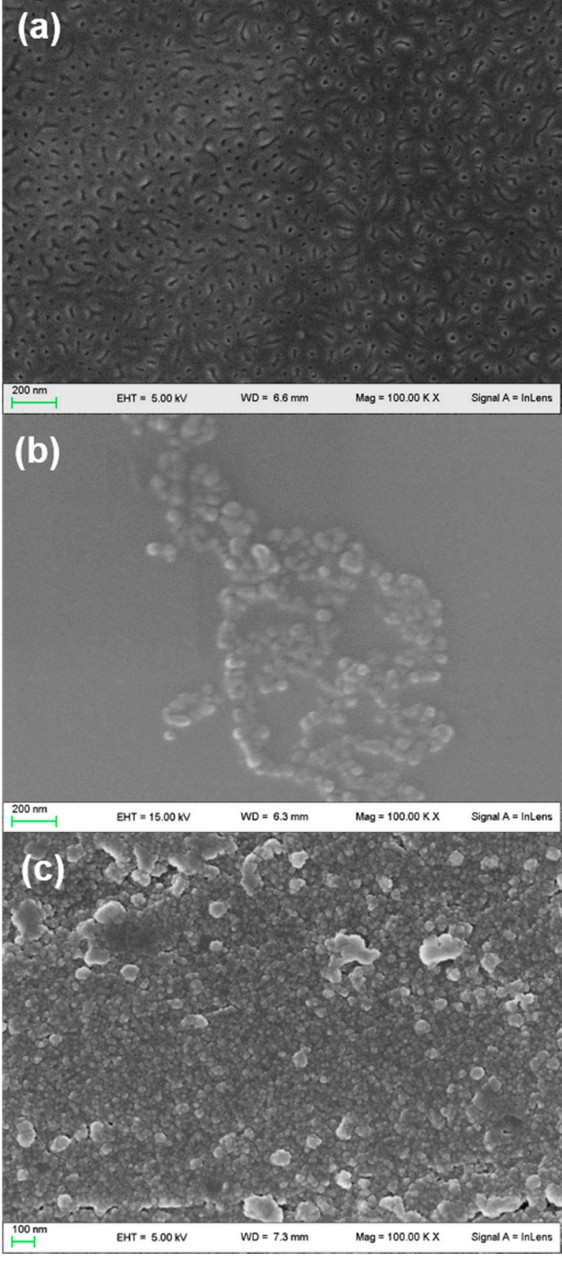

**Figure 1.** SEM images of polylactic acid (PLA) film (**a**), activated PLA film (**b**) and activated PLA film with immobilized enzyme (**c**).

The SEM results have been confirmed by the AFM analysis. Figure 2 shows the AFM topography and the corresponding height profiles of the non-activated PLA film, the activated PLA film and the activated PLA film with immobilized proteases.

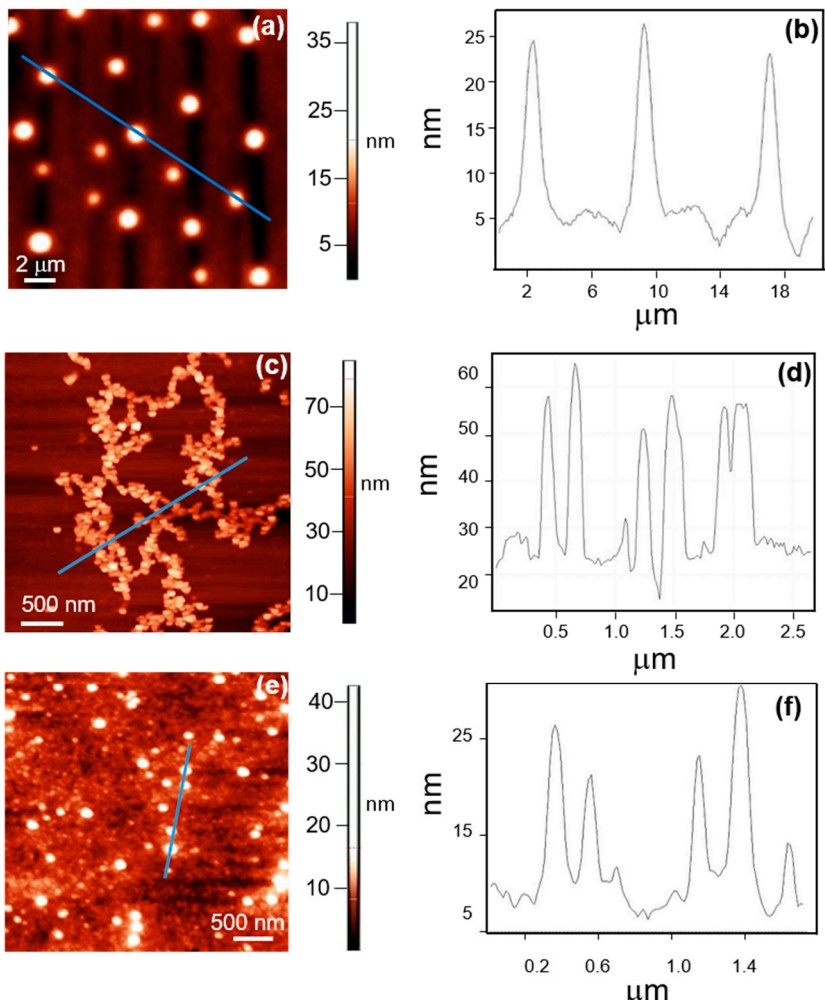

**Figure 2.** (**a**) AFM topography image of the non-activated PLA. (**b**) height profile of the non-activated PLA aggregates observed in the image reported in panel (**a**). (**c**) AFM topography image of activated PLA. (**d**) height profile of the activated PLA aggregates observed in the image reported in panel (**c**). (**e**) AFM topography image of activated PLA with immobilized enzyme. (**f**) height profile of the PLA aggregates observed in the image reported in panel (**e**).

It can be seen that before activation PLA arranges in very large globular aggregates, whose height ranges between 10 and 20 nm, (Figure 2a), while after activation PLA forms irregular aggregates with heights ranging between 20 and 40 nm (Figure 2c).

Vertical profiles of the aggregates of the non-activated and the activated PLA are reported in Figure 2b,d, respectively. After enzyme immobilization, AFM images in Figure 2e,f show globular structures, whose height is about 20–25 nm, completely surrounded by proteases.

Figure 3 shows a comparison between SEM and AFM images of the immobilized proteases and the height distribution of the structures observed by AFM. The protease aggregates exhibit the typical appearance (Figure 3a,b) of protein monolayers [21], arranging in very extensive branching structures with heights found in the range between 2 and 6 nm (Figure 3c).

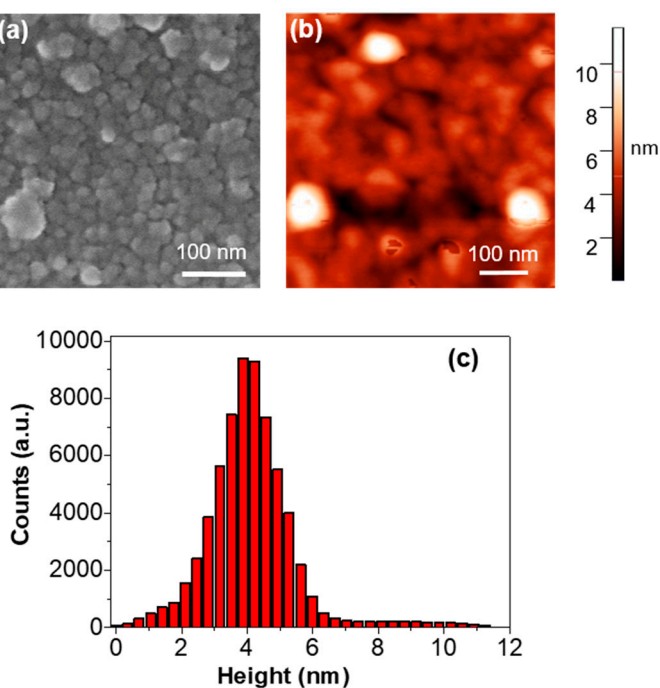

**Figure 3.** Detailed (**a**) SEM and (**b**) AFM topography images of the protease structures. (**c**) histogram showing the height distribution of the protease structures shown in panel (**b**).

*2.2. DSC Analysis*

In order to evaluate the effect of activation and protease treatment, DSC analyses were carried out to study the different thermal properties of produced PLA films. Results from first heating scan are shown in Figure 4 and Table 1.

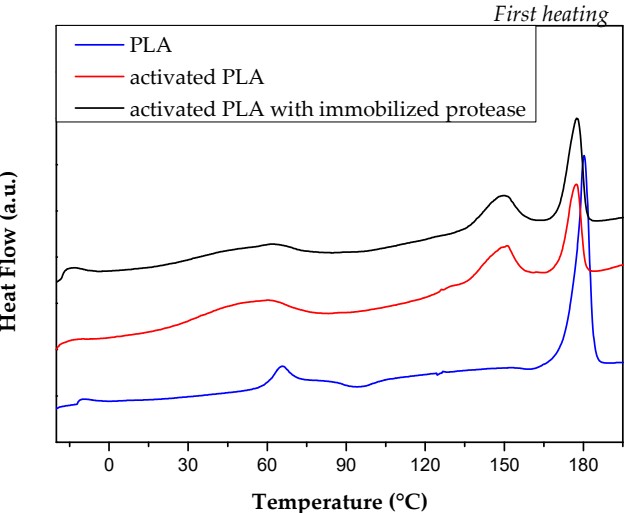

**Figure 4.** Differential scanning calorimetry (DSC) curves of PLA film (**blue line**), activated PLA (**red line**) and activated PLA with immobilized protease (**black line**).

The activation and protease treatment induced some modifications of thermal behavior of neat PLA. Some changes were registered for glass transition, cold crystallization and melting temperatures, as observed from DSC thermograms (Figure 4) and registered thermal parameters (Table 1).

**Table 1.** Thermal properties of the PLA films at first heating scan.

| Formulations | First Heating | | | | | | | |
|---|---|---|---|---|---|---|---|---|
| | $T_g$ (°C) | $\Delta H_{cc1}$ (J/g) | $T_{cc1}$ (°C) | $\Delta H_{cc2}$ (J/g) | $T_{cc2}$ (°C) | $\Delta H_m$ (J/g) | $T_m$ (°C) | $X_m$ (%) |
| PLA | 62.3 | 2.9 | 94.2 | 1.5 | 160.5 | 40.7 | 180.3 | 39.1 |
| activated PLA | - | - | - | - | - | 43.3 | 150.4/177.2 | 46.5 |
| activated PLA with immobilized protease | - | - | - | - | - | 42.2 | 149.6/177.6 | 45.4 |

While glass temperature ($T_g$) for PLA film was easily detected and measured 62.3 °C, the presence of a superimposed endothermic event in the same temperature range prevented to measure the same characteristic temperature for both activated and protease activated PLA films. While neat PLA film shows two cold crystallization peaks centered at $T_{cc1}$ = 94.2 °C and $T_{cc2}$ =160.5 °C ($\Delta H_{cc1}$= 2.9 J/g and $\Delta H_{cc2}$= 1.5 J/g, respectively), cold crystallization events were completely inhibited with the activation process; on the other hand, sensible variations were observed in the melting phenomena. Activated PLA films showed double melting peak temperatures ($T_m$), at two distinct temperatures, that can be related to the melting of different crystals formed during the process and treatment, that can be correlated with the presence of low molecular weight fractions. It is reasonable to think that diethylentriamine (DETA) activating agent interacted with the polymeric fraction, that underwent amino reaction [22]. The increase in crystallinity degree, registered for activated PLA (PLA $X_m$ = 39.1 %, activated PLA $X_m$ = 46.5% and activated PLA with immobilized protease $X_m$ = 45.4%), highlights that the activation process is able to reduce the amorphous component, which is not further modified by the attachment of the enzymes.

### 2.3. Raman Spectroscopy

Raman spectroscopy is a powerful and non-invasive method for the chemical recognition of different species even in complex materials. Thanks to its high specificity, it is widely used in material science [23,24] as well as to analyze complex structures of biological interest [25,26]. In the present case, Raman spectroscopy was used to verify the presence of the immobilized enzyme in the final product. The spectra of the investigated samples are shown in Figure 5. Different structured bands and simple peaks are present in the spectra. Each peak corresponds to a particular vibrational mode of the molecules present inside the scattering volume. In particular, comparing the spectrum of the activated PLA, reported as a red line in Figure 5, with the one of the activated PLA with immobilized enzymes, black line in Figure 5, some peaks are observed in both spectra, namely those centered at about 1450 and 1300 cm$^{-1}$ attributed to the twisting and bending vibrations of the hydrogens in the $CH_2$ group, respectively. The peak around 1600 cm$^{-1}$ attributed to the $NH_2$ bending mode [27,28] of diethylentriamine is, instead, only present in the activated PLA, and it disappears after the enzyme linking with the matrix is established. This feature is the sign of a structural modification brought about by the immobilization process.

On the other hand, the peak at about 1660 cm$^{-1}$ only appears in the spectrum of activated PLA with immobilized protease. It can be attributed to both the C=N and amide 1 vibrational modes peculiar to protein structures. Hence, this peak is the spectroscopic marker of the reaction achievement and it confirms the presence of proteases in the final sample.

The spectral shape of the high frequency part of the Raman spectra is also informative about the sample composition and its evolution. The =CH, $CH_2$ and $CH_3$ stretching vibration bands located between 2800 and 3100 cm$^{-1}$ are particularly sensitive to modifications in protein concentration [26] or to the structural packing of the chemical species [29,30]. In this case, after enzyme immobilization, a strong increase of the peak at 2930 cm$^{-1}$, and a relative decrease of the =C-H vibrational mode at 3050 cm$^{-1}$ is detected. The peak at 2930 cm$^{-1}$ is assigned to the $CH_3$ stretching of proteins [25] and again confirms the protease presence in the final product.

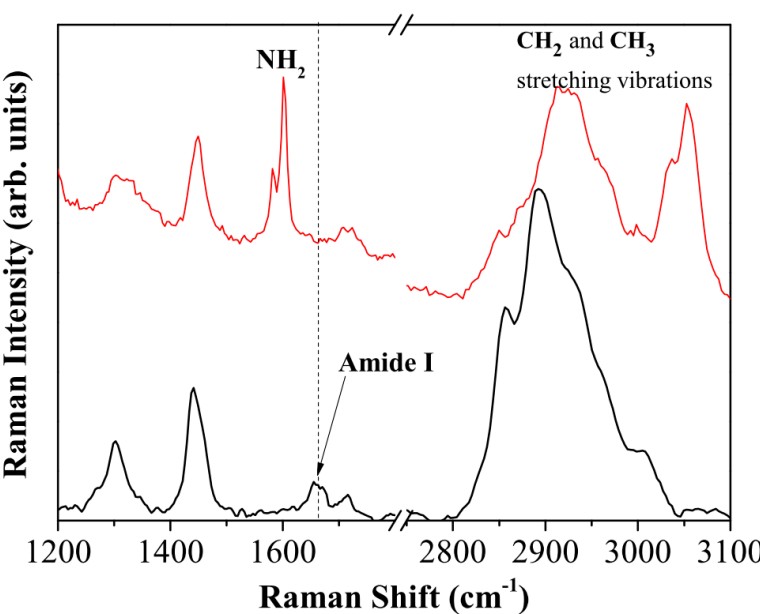

**Figure 5.** Raman spectrum of activated PLA (**red line**) compared with the Raman spectrum of activated PLA with immobilized protease (**black line**).

### 2.4. Stability of Enzymatic Activity over Time

Although the immobilization was proved by Raman spectroscopy, a quantitative assessment of protease immobilization yield was given by the Bradford method, as described in the Material and Methods Section. An average value of 68% immobilization yield, corresponding to 170 μg/cm$^2$, was found. Subsequently, the activity of immobilized proteases was tested repeatedly, using casein as a substrate. The experiments were carried out at a temperature of 55 °C and a pH of 8.6, which were chosen as the optimum conditions based on the studies shown in Section 3.5. The catalytic activity towards casein was checked during more than 20 cycles of hydrolysis, spanning a period of more than six months, while storing the PLA-protease films between each experiment in 50 mM potassium phosphate buffer (pH 8.6) at 4 °C, without needing any particular treatment. Although most of the unbound enzyme had detached after water washes, as assessed by the Bradford assay (see Section 3.2), the first two hydrolysis cycles produced an initial decrease in activity, probably due either to a further detachment of those enzymatic molecules that did not steadily bind to the material or to the loss of activity for some of the protease types present in the cocktail.

As shown in Figure 6a starting from the third cycle, immobilized enzymes retain hydrolytic activity during several following hydrolysis cycles, just slowly declining, ultimately outdoing the performance of free enzyme.

By fitting the activity trend with a linear regression function, one can carry out a valuation of the possible number of cycles the system can undergo before the activity is reduced to zero, which results in about 100 cycles. The enzymatic activity of immobilized proteases adds up at each utilization, exceeding that of free proteases after twelve cycles of hydrolysis (Figure 6b). The total enzymatic activity after estimated 100 cycles is about 460 mU per mg of immobilized enzyme, more than quadrupled with respect to the 110 mU/mg activity of the one-time usable free enzyme.

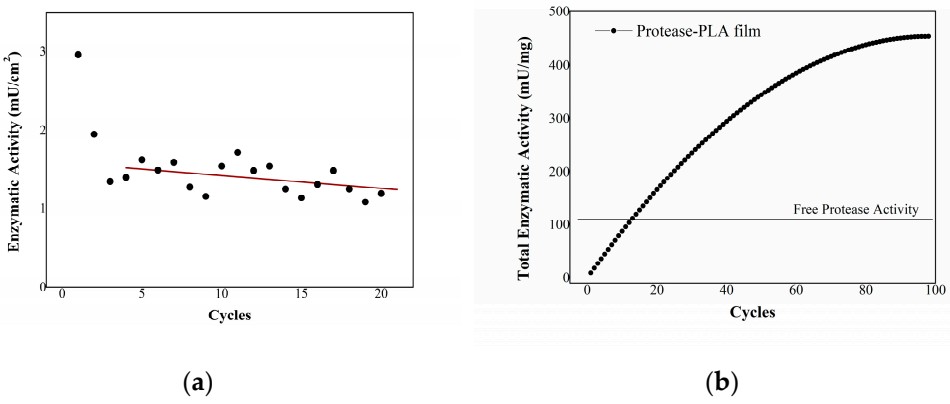

(**a**)                                     (**b**)

**Figure 6.** (**a**) enzymatic activity of immobilized proteases using casein as substrate. In the axis of abscissas, the cycles of hydrolysis are reported. (**b**) total enzymatic activity of protease-PLA films as a function of hydrolysis cycles, compared to free protease activity.

### 2.5. Temperature and pH Effect on Enzymatic Activity

Figure 7 shows the activity values obtained by carrying out the casein test on free (solid line) and immobilized (dashed line) proteases at various temperatures.

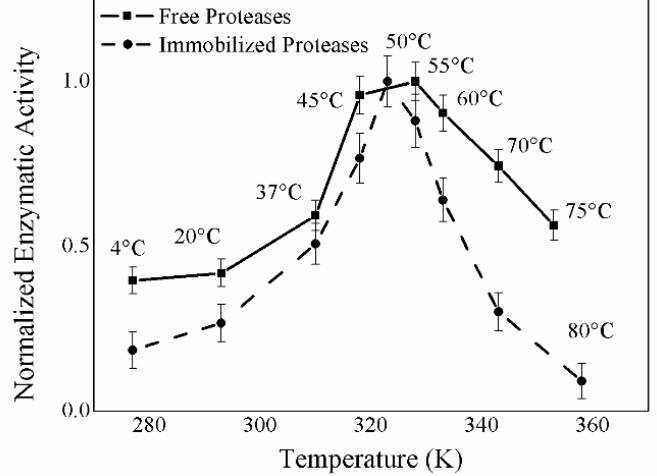

**Figure 7.** Temperature effect on the enzymatic activity of immobilized proteases using casein as substrate. In the axis of abscissas temperature (K) is reported, in the ordinate axis normalized enzymatic activity at different temperatures is reported.

The enzymatic activity of immobilized enzymes exhibits a bell-shaped trend: It is almost zero at low (4 °C) and high temperatures (80 °C) and reaches good values between 37 and 60 °C, with its optimum being at 50 °C. Above 60 °C, the enzymatic activity decreases because of the occurrence of denaturation processes. Figure 7 provides a basis for comparison between free and immobilized proteases, showing that the immobilization process does not alter the temperature optimum, which can still be found between 45 and 55 °C; however, the activity of immobilized proteases decreases more rapidly, moving away from the optimum point, probably because free enzymes are more likely to adapt to changes in temperature. Figure 8 shows the activity values obtained by carrying out the casein test on free (solid line) and immobilized (dashed line) proteases at various pHs. For this purpose, casein was dissolved in potassium phosphate buffer at different pHs. The activity of the free and immobilized proteases at the various pHs shows bell-shaped trends, very much resembling each other, in which under the more stringent conditions of pH 5.0 and 12.2 the enzymes are not active, while they are well-operating at the other pH values, with optimal efficiency exhibited between 7.5 and 10.0.

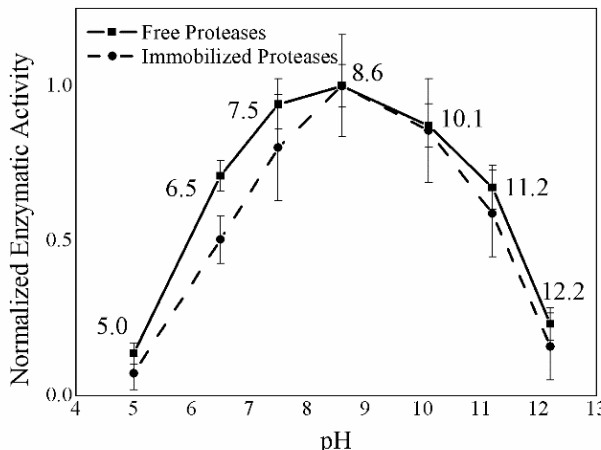

**Figure 8.** pH effect on the enzymatic activity of free (**solid line**) and immobilized proteases (**dashed line**) using casein as substrate. In the axis of pHs values are reported, in the ordinate axis normalized enzymatic activities at different pH are reported.

The temperature and pH effect on the enzymatic activity of immobilized proteases was tested repeatedly at different times after the immobilization, over the course of six months, in order to check the inalterability of the enzyme properties over time. The values collected at a given temperature were comparable in the various detections and the data reported in the graph are the mean values obtained taking into account all the measurements, together with the relative error bars calculated as standard deviations.

*2.6. Hydrolysis of Model Biomasses*

In order to prove the efficacy of the investigated system as a biocatalytic technology for the degradation of biomasses, the hydrolysis performed by immobilized proteases on a simple protein, BSA, was investigated. A 1 mg/mL solution of BSA was put in contact with the enzyme-loaded PLA film, and the result of the degradation checked after 2, 4, 8 and 24 h of incubation at 55 °C. The temperature was chosen based on the temperature effect results, which had shown a peak in the activity of immobilized proteases around this temperature value. The experiment was performed on films of activated PLA with immobilized enzymes, after they had already undergone some cycles of hydrolysis, so that they had reached a plateau in the enzymatic activity value. This made us confident that all the loosely bound molecules had been removed from the film and that all the hydrolytic activity could be assigned to the action of immobilized proteases.

In Figure 9a, where the results of the SDS-PAGE experiment are reported, a band corresponding to BSA (MW = 66.5 kDa) is readily detectable. As it is clear from the reduced intensity of the band along the horizontal lane (shown in the bar graph in Figure 9b), a partial degradation of the biomass operated by immobilized proteases is revealed during the first hours of incubation. The hydrolysis becomes significant after 8 h of incubation, when the intensity of the band is reduced down to almost 40% of its initial value, and it is almost complete after 24 h: In fact, in the last lane of the SDS-PAGE, the apparent bleaching of the band assigned to BSA (whose intensity becomes less than 10%) is the sign of the quantitative degradation of the biomass performed by immobilized proteases. Moreover, the absence of the protein pattern peculiar to free proteases (right-hand lane in Figure 9a) confirms that no protease detachment occurs during the experiment; hence, the proteolytic activity toward BSA is exclusively performed by immobilized enzymes.

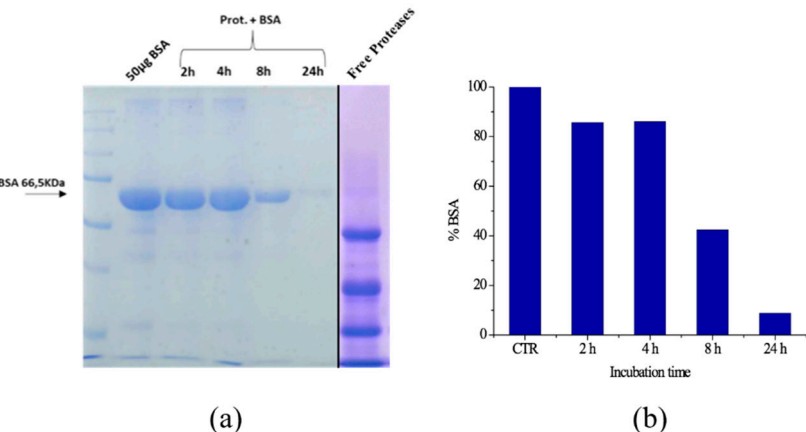

**Figure 9.** Bovine serum albumin (BSA) hydrolysis after 2, 4, 8 and 24 h at 55 °C evaluated by SDS-PAGE electrophoresis and free proteases, left image (**a**) and histogram showing the BSA band intensity reduction by increasing incubation time, evaluated with ImageJ, right graph (**b**).

Subsequently, the possibility of reusing the immobilized proteases several times has been verified using as an example of waste biomass a protein extract of soybeans obtained as described in the materials and methods section. A series of ten hydrolysis cycles was conducted on the same protease-loaded PLA film. In each experiment, soybean waste extracts (1 mg/mL protein content) were kept in contact with the immobilized proteases, then aliquots were taken at different times and their hydrolysis degree again evaluated by SDS-PAGE electrophoresis. Between one experiment and the following, the film was washed and put in contact with a new soy extract. As representative examples, Figures 10 and 11 show the results of two of the ten hydrolysis cycles conducted on the same protease-loaded PLA film. In particular, Figure 9 shows the result of the fifth hydrolysis cycle, while Figure 11 refers to the eighth cycle. The soy extract, as it is shown in the control (CTR) lane of the gel in Figures 10a and 11a, features a variegated protein pattern with two main bands, whose molecular weights are about 35 and 13 kDa. As it is readily visible from the gel, these bands rapidly reduce in intensities with increasing time. From a quantitative point of view, the densitometry analysis of the 35 kDa band (Figures 10b and 11b) reveals a high degree of digestion already after the first 30 min of incubation, when its degradation is about 75% of the initial protein content in both cases. In the first experiment (fifth cycle), after four hours, the hydrolysis can be considered almost quantitative, with only 5% of the original soy protein still to be digested. This value is further reduced to 3% after overnight incubation.

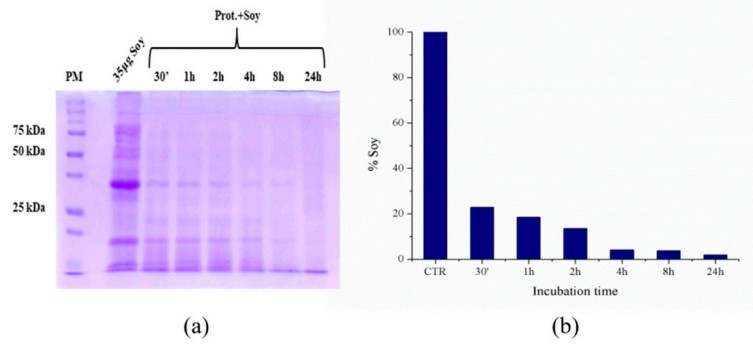

**Figure 10.** Soy waste extract hydrolysis (fifth hydrolysis cycle) after 30′, 1, 2, 4, 8 and 24 h at 55 °C evaluated by SDS-PAGE electrophoresis, left image (**a**) and histogram showing the soy band intensity reduction by increasing incubation time, evaluated with ImageJ, right graph (**b**).

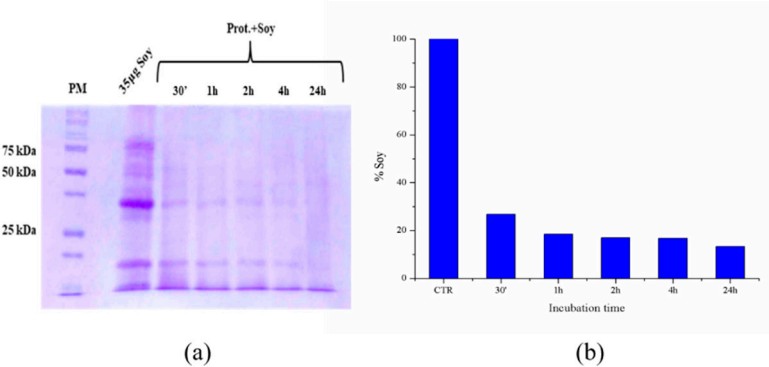

(a)           (b)

**Figure 11.** Soy waste extract hydrolysis (eighth hydrolysis cycle) after 30′, 1, 2, 4, 8 and 24 h at 55 °C evaluated by SDS-PAGE electrophoresis, left image (**a**) and histogram showing the soy band intensity reduction by increasing incubation time, evaluated with ImageJ, right graph (**b**).

As is the case with the second experiment (eighth cycle), a good degree of hydrolysis is again observed after 24 h of incubation, when the undigested biomass is nearly 10%, showing the great potential of our reusable PLA-protease system.

### 2.7. pH-Stat

With the aim being to have a quantitative evaluation of the DH promoted by immobilized proteases, the pH-stat method was employed on protein extracts of soybeans placed in contact with the protease-loaded PLA films at 37 °C and pH = 7.5. The pH-stat technique allows to follow the DH over time monitoring the number of peptide bonds cleaved by the enzymes. By plotting the DH value as a function of time, a hydrolysis curve is obtained as reported in Figure 12. The plot shows an initial quasi-linear trend during the first five hours of operation, later reaching a plateau in about eight hours, where the DH is recorded as 37% of the total peptide bonds.

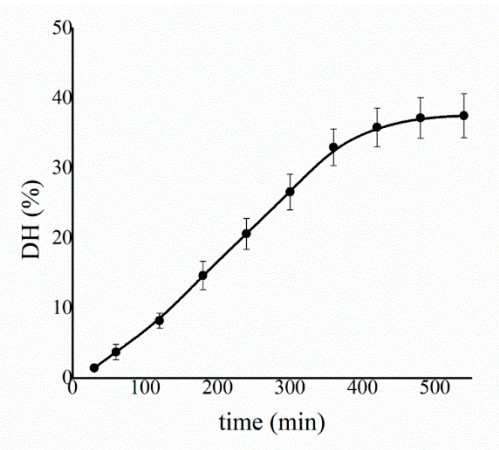

**Figure 12.** Soy extract degree of hydrolysis (DH) determined by the pH-stat method as a function of time at 37 °C and pH = 7.5. The results are average values collected during five independent experiments.

### 2.8. Peptide Analysis in the Protein Hydrolysate

The soybean protein hydrolysate obtained after the hydrolysis performed by immobilized proteases was analyzed by LC/MS. The control, consisting of a raw extract of soybean waste, and the hydrolysates produced after 24-h hydrolysis were searched for tri- and tetrapeptides with the aid of the MassHunter Metlin Peptides AM PCD database. Figure 13a shows a graph reporting the abundance, measured as the area under the chromatogram peaks, of the total peptides detected in the hydrolysates, compared with the

control. The presence of some peptides already in the control extract may be due to the thermal treatment of the initial soybean powder, carried at 80 °C as described in the Material and Methods section. However, in the 24-h hydrolysate, a massive increase of total peptides is detected, thus corroborating the results acquired with the SDS-PAGE and pH-stat method and confirming the efficient functioning of protease-loaded PLA films. The abundance of the identified peptides is shown in Figure 13b, where the most abundant species found is the tetrapeptide Ala-His-Ile-Ser, followed by two other tetrapeptides, namely Ala-His-Pro-Thr and Asp-Pro-Pro-Val. As is well-known for protein hydrolysates, these peptides arouse interest for their potential biological activity. In fact, small peptides derived from the hydrolysis of soy have previously been found to exhibit antioxidant activity [31]. In particular, the antioxidant effects of histidine-containing peptides might be associated to their ability to act as metal chelator, oxygen quencher and scavenger of hydroxyl radicals [32]. Moreover, as is the case with the detected Asp-Pro-Pro-Val peptide, the hydrophobic valine amino acid at the N-terminal position together with the presence of proline in the sequence has been related to antioxidant activity, as well [31]. However, the biological effects of the soy-hydrolysates obtained through our PLA-protease system will be the object of future investigations.

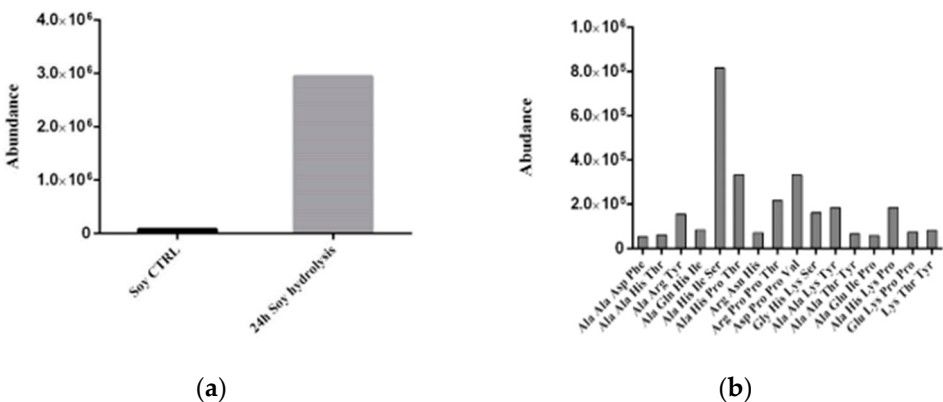

(**a**)                                                                                                   (**b**)

**Figure 13.** (**a**) abundance of total peptides in control extracts and 24-h soy hydrolysates. (**b**) types and abundance of peptides detected in the hydrolysate from soy wastes after 24-h treatment with protease-loaded PLA films.

## 3. Materials and Methods

### 3.1. Materials

PLA (average Mn = 40,000 g/mol), diethylentriamine, 25% glutaraldehyde stock solution and proteases from *Aspergillus oryzae* were purchased from Sigma-Aldrich (3050 Spruce St, Saint Louis, MO, USA) and used without any further purification. These fungal proteases consist of a cocktail of different enzymes exhibiting both endoprotease and exopeptidase activities.

For Casein Assay, casein from bovine milk, Folin and Ciocalteu's phenol reagent, trichloroacetic acid purchased from Sigma-Aldrich (3050 Spruce St, Saint Louis, MO, USA), sodium carbonate anhydrous from Carlo Erba (Chaussée du Vexin, 27100, France) and potassium phosphate di-basic from Panreac (Polígono Pla de la Bruguera E-08211 Castellar del Vallès, Barcelona, Spain) were used. For sodium dodecyl sulphate-polyacrylamide gel electrophoresis (SDS-PAGE) electrophoresis 30% acrylamide/bis solution purchased from BIO-RAD (1000 Alfred Nobel Drive. Hercules, 94547, CA, USA), trizma base, glycine reagent, *N,N,N,N′*-tetramethylethylenediamine and ammonium persulfate from Sigma-Aldrich (3050 Spruce St, Saint Louis, MO, USA) and SDS pure from Panreac (Polígono Pla de la Bruguera E-08211 Castellar del Vallès, Barcelona, Spain) were employed. For coloring gels, different solutions of methanol and ethanol absolute purchased from VWR Chemicals (100 Matsonford Rd, Radnor, 19087, PA, USA), acetic acid glacial from Carlo Erba (Chaussée du Vexin, 27100, France) and Coomassie Brilliant Blue from BIO-RAD

(1000 Alfred Nobel Drive. Hercules, 94547, CA, USA) were used. For testing the hydrolytic activity of immobilized proteases, BSA purchased from Sigma-Aldrich was employed, as well as a soybean extract obtained from waste soybean seeds provided by Società Agricola Cave Mevania di Cauli Loredana e Rosati Filippo and C. SS (Via Della Rotta 25-06034 Foligno, Perugia, ITALY).

### 3.2. Enzyme Immobilization

The PLA films were prepared by means of a spin coating technique: 0.25 g of PLA powder were dissolved in 5 mL of chloroform. Part of the solution was deposited on a polytetrafluoroethylene plate which was placed in the rotor of a centrifuge at 3000 rpm for 1 min. The material was left to dry for 48 h at room temperature. After that, PLA films needed to be activated: The activation protocol involves the treatment of the material with 21% diethylenetriamine for 1 h at 55 °C. After abundant washes with deionized water, the material was processed with a 2.5% glutaraldehyde solution for 3 h at room temperature. Finally, the material was washed with water and then the enzyme solution (500 μL) containing 250 μg proteases was placed on 1 cm$^2$ films and left to bind overnight. This amount of protease was chosen after testing different concentrations of the enzyme, in that higher values supposedly exceed the capacity of the matrix, thus leading to the loss of most proteases as unbound enzymes. The immobilization yield was determined by the Bradford method [33], using Coomassie Brilliant Blue reagent and measuring the absorbance at 595 nm. The percentage of bound protein was indirectly determined as the difference between the amount of proteins in the starting solution and those retrieved in the solution collected after the immobilization and in the water volumes used to wash the material (three washings with 500 μL each time).

### 3.3. Morphological and Chemical Characterization

A correlative analysis has been performed to characterize the micro and nano morphology of the samples by SEM and AFM studies. For the SEM analysis three samples were prepared: The first consists of a film of PLA, without further treatment. The second is a PLA film functionalized with diethylenetriamine and activated with glutaraldehyde. The third sample is the activated PLA with the immobilized enzymes. The same three samples were deposited on freshly cleaved ruby mica for the AFM analysis. SEM images were obtained using a Field Emission Gun Electron Scanning Microscopy LEO 1525 (ZEISS, Jena, Germany), after metallization with graphite. The images were acquired by an Inlens detector (ZEISS, Jena, Germany). Tapping mode AFM measurements were carried out in air using a Solver Pro scanning probe microscope (NT-MDT, Moscow, Russia). Rectangular silicon cantilevers, 100 μm long, having typical resonance frequency and spring constant 255 kHz and 11.5 N/m, respectively, were used [34].

Differential scanning calorimeter (TA Instrument, Q200, New Castle, Delaware, United States) investigations were performed from −25 to 210 °C, at 10 °C min$^{-1}$, applying two heating and one cooling scans, in nitrogen atmosphere (50 mL min$^{-1}$). Melting and cold crystallization temperatures and enthalpies ($T_m$, $T_{cc}$ and $\Delta H_m$, $\Delta H_{cc}$) were determined from the first heating scan. The glass transition temperature ($T_g$) was registered from the first heating scan.

The crystallinity degree was calculated as Equation (1):

$$X = \frac{1}{1 - m_f}\left[\frac{\Delta H_m - \Delta H_{cc}}{\Delta H_0}\right] * 100 \tag{1}$$

where $\Delta H$ is the enthalpy for melting ($\Delta H_m$) or cold crystallization ($\Delta H_{cc}$) and $\Delta H_0$ is the enthalpy of melting for a 100% crystalline PLA sample, taken as 93 J g$^{-1}$ [35] and $(1 - m_f)$ is the weight fraction of PLA in the sample.

The biochemical characterization of the activated PLA and of the activated PLA with immobilized enzyme, was performed by Raman spectroscopy. In this case the samples were deposited on silicon slabs to reduce the Raman signal from the substrate. The micro-Raman

setup consists of an iHR320 Triax Imaging Spectrometer of the HORIBA Jobin Yvon (Kyoto, Japan) coupled in a custom configuration with a CM1 microscope purchased form the JRS Scientific Instruments (Mettmenstetten, Switzerland) [36]. A solid-state laser operating at λ = 532 nm was used, with laser power reduced to less than 10 mW to decrease the thermal effects on the sample. The microscope was equipped with a Mitutoyo M-Plan Apo 20× with a very long working distance (20 mm) and with numerical aperture (NA) of 0.42, providing a lateral resolution of about 3 μm. The Raman signal collected in back scattering configuration was dispersed by a 600 grooves/mm grating and acquired by a CCD detector (1024 by 256 pixels), allowing the simultaneous acquisition of the Raman shift in the 100–3500 cm$^{-1}$ spectral range with a resolution of about 10 cm$^{-1}$. Different points were analyzed for each sample to takes into account the sample heterogeneity. The luminescence background was fitted by a spline function and subtracted from the spectra [25].

### 3.4. Enzymatic Tests

The proteolytic activity of immobilized proteases was evaluated using the Anson's and Folin and Ciocalteu's methods [37,38] employing casein as substrate. A 0.65% weight/volume casein solution, prepared by dissolving 6.5 mg per each mL of a 50 mM potassium phosphate buffer at pH 8.6, was used. The solution temperature was gradually increased with gentle stirring up to 80 °C for 10 min. Then, 500 μL of casein solution was put with the immobilized enzymes at 55 °C for 30 min or 1 h depending on the sample. During this incubation time, the protease activity promotes the liberation of tyrosine. After the incubation time, the casein solution was collected in a vial and the reaction was stopped by adding 500 μL of a 110 mM trichloroacetic (TCA) acid solution, then leaving the vials at 55 °C for 30 min. After centrifugation, 500 μL of the casein–TCA supernatant solution were taken into another vial where 1250 μL of 0.5 M a sodium carbonate solution and 250 μL of Folin and Ciocalteu's phenol reagent were added. The vials were mixed and then incubated at 37°C for 30 min; after that, the absorbance values of the solution were recorded at 750 nm.

The absorbance values are proportional to the activity of the immobilized proteases. These data are then compared to a standard curve obtained from the reaction of Folin and Ciocalteu's phenol reagent with known quantities of tyrosine. From the standard curve, the micromoles of tyrosine equivalents released are determined, and then the activity of immobilized proteases can be calculated in terms of milliunits per square centimeter (mU/cm$^2$), by using the following equation (Equation (2)):

$$Activity \left( \frac{mU}{cm^2} \right) = \frac{mol\ tyrosine\ equivalents\ realesed\ (nmol)}{time\ (min)\ \times\ area\ (cm^2)} \tag{2}$$

where 1 mU of protease activity is defined as the quantity of enzyme needed to hydrolyze casein to 1 nmol of tyrosine in 1 min [39]. The catalytic activity of immobilized proteases towards casein was measured over time, by carrying out one measurement of enzymatic activity per week. Between one measurement and the other the protease-loaded PLA films were kept in saline buffer, pH 7.5 at 4 °C under sterile conditions. The temperature effect on the enzymatic activity of immobilized proteases was evaluated using the same casein assay, performing the first incubation step at the following temperatures: 4, 20, 37, 45, 55, 65 and 80 °C, obtained by thermostating the system at the desired temperature. Conversely, in order to test the pH effect, the same assay was carried out by dissolving casein in buffered solution in the pH range from 5.0 to 12.2.

### 3.5. Protein Extract from Soybean Waste

Soy is one of the most cultivated products in the world; in fact, it is a food rich in proteins, polyunsaturated lipids and glucosides. An important portion of the world soybean production is destined to human consumption but also to that of farmed animals in the form of flours and cakes. In this work soybean seed wastes derived from the industrial processing of this product were used. The waste seeds were mechanically shredded until a

homogeneous powder was obtained. The soy powder was then processed for the extraction of the total proteins to be used as substrates to test the hydrolytic activity of the proteases immobilized on the PLA column. The soy powder was suspended in deionized water and incubated for 1 h at 80 °C. During the incubation, the sample was repeatedly shaken to favor the extraction of proteins. At the end of the incubation time, the sample was centrifuged at 16,000× *g* and 4 °C for 15 min and the soluble part was collected. This solution was further centrifuged at 16,000× *g* for 15 min and the supernatant containing the total extract of solubilized proteins was quantified by the Bradford protein assay and then used as a substrate for enzymatic hydrolysis reactions.

### 3.6. Polyacrylamide Gel Electrophoresis in Denaturing Conditions (SDS-PAGE)

The hydrolysis of 1 mg/mL solutions of bovine serum albumin and soybean protein extracts was evaluated through SDS-PAGE, which was conducted according to the Laemmli's method [40] using a 12% (*v/w*) acrylamide separating gel and a 4% acrylamide stacking gel. Samples were prepared by mixing them with 5× sample buffer (0.5 M Tris–HCl buffer pH 6.8, containing 10% (*w/v*) SDS, 50% (*v/v*) glycerol, 0.01% (*w/v*) bromophenol blue and 125 mM dithiothreitol, DTT) and incubated at 95 °C for 5 min. Subsequently, appropriate volumes were loaded into the gel (corresponding to 50 and 35 µg of initial proteins for BSA and soybean extracts, respectively) and subjected to a constant electrophoretic run at 40 mA. The stroke was carried out using the known molecular weight standard as a reference. The running buffer used (electrode buffer) was Tris 0.025 M/glycine 0.192 M containing 1% SDS (*w/v*).

### 3.7. Coomassie Brilliant Blue Gel Staining

The gels obtained were colored by Coomassie Brilliant) staining to highlight the protein banding of the various samples. Coomassie Brilliant Blue staining is based on the non-specific binding of Coomassie Blue R250 protein-dye. Before being colored, the gel was immersed in a solution of 40% methanol and 10% acetic acid to fix the proteins inside the mesh of the gel (fixing solution). The gel was then incubated for 90 min in a 0.02% solution of Coomassie Brilliant Blue R-250 and finally washed for 4 h with a solution of 25% ethanol and 8% acetic acid to remove the dye that does not bind to the proteins and diffuses in the gel (destaining solution). The band intensities in the stained gels were finally quantified with the ImageJ software (Madison, WI, USA).

### 3.8. pH-Stat

The degree of hydrolysis (DH) of soy waste extracts was monitored over time with the pH-stat technique [41,42]. Starting from a mildly alkaline pH, the hydrolytic reaction is responsible for pH reduction, which can be compensated by adding specific amounts of base (NaOH). By doing so, the hydrolysis equivalents can be calculated as follows (Equation (3)):

$$h = B \times N_b \times \frac{1}{\alpha} \times \frac{1}{MP} \tag{3}$$

where $B$ is the volume of base consumed (mL), $N_b$ is the base normal concentration, $\alpha$ is the average degree of dissociation of the $\alpha$-NH$_2$ groups of amino acids and $MP$ is the mass of soy proteins in the experiment.

In particular, the $\alpha$-NH$_2$ group degree of dissociation is (Equation (4)):

$$\alpha = \frac{10^{pH-pK}}{1 + 10^{pH-pK}} \tag{4}$$

where $pK$ is the average $pK$ of the $\alpha$-NH$_2$ groups released during hydrolysis ($pK$ = 7.5 at 25 °C).

Finally, the DH value can be estimated from the hydrolysis equivalents through the following relation (Equation (5)):

$$DH = \frac{h}{h_{tot}} \times 100 \tag{5}$$

with $h_{tot}$ being the total number of peptide bonds in a protein expressed as meqv g$^{-1}$ of protein). In the case of soy extracts, $h_{tot}$ is known to be 7.8 meqv g$^{-1}$ [41].

*3.9. Peptides Extraction and LC/MS Analysis*

The control, consisting of a raw extract of soybean waste, and the hydrolysates produced after 24-h hydrolysis were analyzed for peptide quantification and identification with LC/MS. Protein hydrolysates obtained from the hydrolysis of the soy waste biomass were diluted with methanol (up to 90% of the final volume) to allow both peptide extraction and protein precipitation. After a centrifugation step, an aliquot of supernatant was subject to LC/MS analysis. LC separation was performed on an Agilent 1260 Infinity LC System (5301 Stevens Creek Blvd—95051 Santa Clara, CA, USA) with a 10 min gradient time on a reverse phase column (Ascentis Express Peptide ES-C18 750 × 2.1 mm, 2.7 um, Supelco, Bellefonte, PA, USA) at 50 °C and 0.5 mL/min flow. The mobile phase consisted of water and acetonitrile, both containing 0.1% formic acid. Positive polarity data were acquired on the Agilent 6530 LC/QTOF using an Agilent JetStream source (5301 Stevens Creek Blvd—95051 Santa Clara, CA, USA) in the range 50–1700 m/z and in AutoMSMS modality, at 5 spectra/"s" and 3 spectra/"s" for MS and MS/MS respectively. The acquired raw data were processed with Agilent MassHunter Bioconfirm Software (5301 Stevens Creek Blvd—95051 Santa Clara, CA, USA) (B.09.00) and searched for the identification of small peptides using MassHunter Metlin Peptides AM PCD library.

## 4. Conclusions

In this work, the possibility of using immobilized enzymes on inert and biocompatible materials for biomass degradation has been demonstrated. Fungus-derived proteases have been covalently bound to PLA, by chemical activation of the polymeric films. DSC analysis has provided a thermal characterization of the produced PLA films, showing an increase in crystallinity after the activation process, while their morphology has been described thanks to SEM and AFM analyses. Protein determination by Bradford method showed an immobilization yield of about 70%. The covalent binding between activated PLA and proteases was proved by Raman spectroscopy and enzymatic activity assays, which revealed a wide range of working pHs, and an optimum temperature of 50 °C. Although an initial loss of activity was detected during the first cycles of hydrolysis, due to the detachment of loosely bound molecules, or partial inactivation of some of the proteolytic enzymes comprising the employed cocktail, the system proved to be stable over a long period of time (exceeding six months) and for over 20 successive operation cycles, during which proteases preserved their activity towards the hydrolysis of model biomasses (casein, BSA and soy waste extract). By doing so, immobilized proteases exhibit a total activity which is significantly higher than that of one-time usable free proteases.

In particular, the results of the soy waste extract hydrolysis are most encouraging in that significant protein degradation is reached already after only 30 min of incubation time, as shown by the SDS-PAGE results, while a high degree of hydrolysis is obtained in eight hours (DH = 37% of total peptide bonds), as demonstrated by the pH-stat analysis; moreover, the experiments have been repeated several times demonstrating how the system can be reused with only a minor reduction in its performance. Furthermore, the direct outcomes of such enzymatic hydrolysis are biologically active low molecular weight peptides featuring numerous advantages in various application areas, whose abundance and composition have been revealed by the LC/MS analysis. These hydrolysates can for example be exploited for their antioxidant effects or used as fertilizers and bio-stimulants in agriculture, by-passing the problem of toxic side products that are formed with the

classic chemical hydrolysis [43]. This study described the great potential of immobilized enzymes, even though the efficiency of the immobilization process should be boosted in order to enhance the performance of the system under investigation. However, being PLA an eco-friendly biomaterial, the described process of protein content recovery and valorization from waste biomass becomes completely green and sustainable. Moreover, the PLA polymer features interesting properties, such as the possibility of being shaped in practically any form. In this light, the synthesis of porous PLA with immobilized proteases could be employed to create a column reactor for the degradation of biomasses in a continuous way. This represents a new enabling green technology for the disposal of agribusiness and food industry wastes and production of high-value bio-based products.

## 5. Patents

This paper is object of an Italian Patent Application n. 102019000025012 filed on 20/12/2019.

**Author Contributions:** Conceptualization, E.C., A.C., D.F., C.E., A.D.M.; methodology, E.C., A.C., D.F., C.E., A.D.M.; formal analysis E.C., A.C., D.F., C.E., A.D.M.; investigation E.C., A.C., S.T., S.C., R.M.P., F.L., A.D.M.; resources, A.D.M., F.C., D.F., C.E.; data curation, E.C., A.C., S.T., S.C., R.M.P., F.L., D.F., C.E., A.D.M.; writing—original draft preparation, E.C., A.C., S.T., S.C., F.L., D.F., C.E., A.D.M.; writing—review and editing, E.C., A.C., D.F., C.E., F.C., A.D.M.; supervision, D.F., C.E.; project administration, D.F., C.E.; funding acquisition, D.F., C.E., F.C., A.D.M. All authors have read and agreed to the published version of the manuscript.

**Funding:** This research received no external funding.

**Data Availability Statement:** This study not report any data.

**Conflicts of Interest:** There are no conflict to declare.

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
