# Peer review of "Covalent Immobilization of Proteases on Polylactic Acid for Proteins Hydrolysis and Waste Biomass Protein Content Valorization"

_catalysts, doi:10.3390/catal11020167_

Round 1

Reviewer 1 Report

Please find my comments below:

  • I would ask the authors to comment on the information contained in the publication and their own research. I note that short peptides containing proline and histidine have toxic effects.
  • The work was not written in a homogeneous language. Please standardise the style.
  • We write the names of chemical compounds with lowercase letters.
  • We write the names of microorganisms in italics.
  • Please standardize the abbreviation of the units. It should be mL, not ml.
  • Linia 419: What did the authors mean when they wrote: „for some of the protease types present in the cocktail.”?
  • Please change the caption for Figure 1.

Author Response

Response to Reviewer 1 Comments

R: I would ask the authors to comment on the information contained in the publication and their own research. I note that short peptides containing proline and histidine have toxic effects.

A: We thank the Reviewer for their comment that pushed us to deeper analyze the results of our research. We added a paragraph about the potential biological effects of small peptides derived from soy hydrolysis, although the thorough investigation of their possible applications will be the object of future studies. However, we have not found any study in the literature claiming that short peptides containing proline and histidine have toxic effects, instead they have previously been found to exhibit antioxidant activity.

R: The work was not written in a homogeneous language. Please standardise the style.

A: The spelling of some words has been modified to homogenize the style to American English.

R: We write the names of chemical compounds with lowercase letters.

A: Chemical compounds are now written with lowercase letters.

R: We write the names of microorganisms in italics.

A: The names of microorganisms are now all in italics.

R: Please standardize the abbreviation of the units. It should be mL, not ml.

A: We corrected the abbreviation of the units.

R: Linia 419: What did the authors mean when they wrote: „for some of the protease types present in the cocktail.”?

A: We apologize with the Reviewer. The sentence was unclear as we did not explain that the proteases we used in our experiments are indeed a cocktail of different endoproteases and exopeptidases. A sentence was added in the Materials paragraph to clarify this point.

R: Please change the caption for Figure 1.

A: The caption of Figure 1 was changed

Reviewer 2 Report

In this paper, proteases from Aspergillus oryzae were immobilized on polylactic acid (PLA) films. The immobilization properties were detailedly characterized by DSC, AFM, SEM, and Raman measurements. The immobilized proteases were stable for over 20 successive operation cycles, exhibiting a total activity which is significantly higher than that of one-time usable free proteases. When soy waste extracts were used as model waste biomass, a high degree of hydrolysis was observed with yielding potentially useful peptides. This work is well designed, clearly described, and rationally interpreted, with demonstrating that fungus-derived proteases covalently immobilized on PLA films would be applicable for hydrolyzing the protein content in waste biomasses. Therefore, this paper is considered acceptable for publishing in its current state.

Author Response

No Response to Reviewer 2